# Multi-view Consistent Image Generation through Self-calibrated Latent Refinement

## Abstract

In this paper, we introduce a novel 3D-aware image generation framework that ensures high-quality and view-consistent image generation. Our core idea is to leverage the semantic latent space of a pre-trained 2D GAN for 3D view-consistent image generation, eliminating need for large-scale dataset use and prior knowledge of camera poses. To achieve this, we propose a latent refiner with multi-view and geometry-preserving capabilities, enabled by self-calibrated depth and pose estimation. Thanks to the advances of diffusion models, our refiner allows for view-consistent latent manipulation in GANs and can be trained using a self-supervised fashion. Our method optimizes the latent codes of a pre-trained 2D GAN across a wide range of pose angles. We demonstrate the effectiveness of our method through evaluations and comparisons with existing baselines on benchmark datasets. Experimental results show the superiority of our method over existing works in both the quality and view-consistency of generated images.

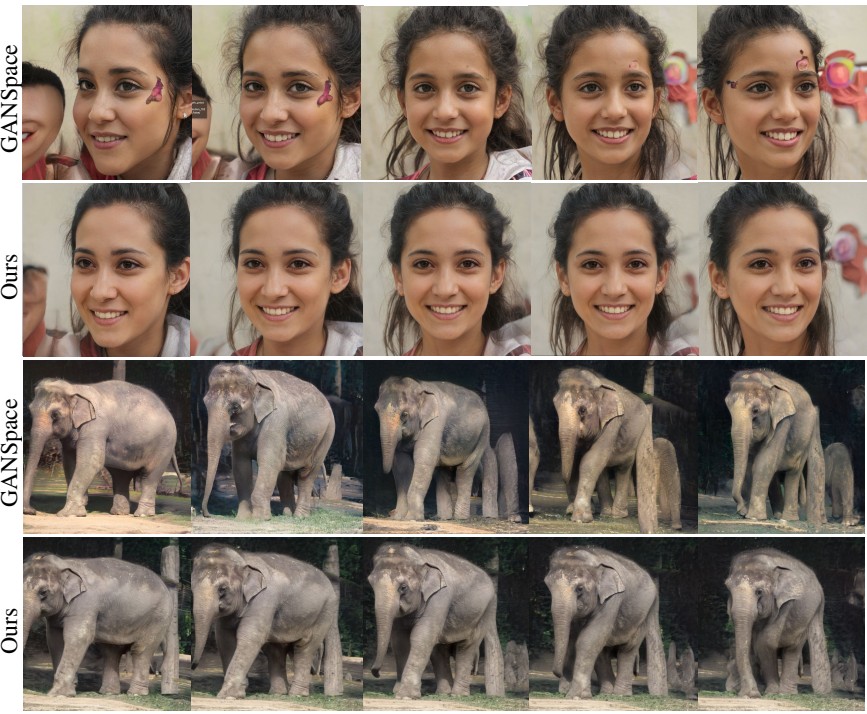

Figure 1: Multi-view image generation from GANSpace (Härkönen et al., 2020) and our method. As shown, direct manipulations on the GANSpace can cause view inconsistency. In contrast, our method, with the proposed latent refiner, achieves seamless rendering results under different rotation angles while maintaining the same object identity.

# 1 INTRODUCTION

Automatic 3D asset generation has witnessed rapid developments in recent years. Such progress has empowered the generation ability of neural networks in a wide range of data generation-related tasks such as VR/AR applications and interactive games. Recently, with the advancement of generative models and 3D representation, 3D-aware GANs (Chan et al., 2022; Skorokhodov et al., 2022; Schwarz et al., 2022; Gu et al., 2022; Deng et al., 2022) and conditional 3D diffusion models using text and images (Liu et al., 2024; 2023b; Long et al., 2023; Qian et al., 2024) have made significant progress with potential applications across various industries adopting visual computing methods.

Among these generative models, we are inspired by 3D-aware GANs (Chan et al., 2022) that extends a 2D GAN by incorporating neural implicit representations and rendering, resulting in improved multi-view consistency. However, 3D-aware GANs require extensive training data to be able to maintain the quality of generated images. These methods necessitate significant computational resources, particularly during rendering and training phases, leading to longer training time and need for high-performance GPUs. Computational constraints often mandate lower-resolution rendering, followed by 2D upscaling, which can compromise view consistency. GMPI (Zhao et al., 2022) and Ray-conditioning (Chen et al., 2023) suggest another approach to reduce the computational issues. Specifically, GMPI modifies a 2D GAN to generate multi-plane images, ensuring view consistency and 3D awareness with minimal structural changes. Ray-conditioning (Chen et al., 2023) enhances photo-realism over photo-consistency, conditioning 2D GANs with light-field priors to produce high-resolution images without using comprehensive 3D models. However, they still require pose distribution and large-scale data sources. Additionally, their ability to maintain 3D consistency may diminish at wider viewing angles.

In this paper, we opt to investigate pre-trained 2D GANs and adapt them for 3D-aware image synthesis. Our motivation for this choice is to avoid any dependency on use of large-scale multi-view or 3D data in training as these are usually difficult and expensive to be obtained. We also choose GANs instead of diffusion models (Ho et al., 2020) as, compared with diffusion models, by design, GANs' latent spaces are more interpretable and can be computed in much faster speed. Furthermore, latent manipulation in a 2D GAN (Shen et al., 2020; Shen & Zhou, 2021; Härkönen et al., 2020; Wu et al., 2021; Zhu et al., 2023; Tewari et al., 2020) has demonstrated an ability to learn 3D-aware features, resulting in generated images with diverse object poses. Inspired by the ability to learn implicit 3D knowledge in 2D GANs, our goal is to generate multi-view consistent images by refining the latent space of a pre-trained 2D GAN with a tailor-designed latent refiner, enabling cross-view and identity consistency (see Fig. 1). Note that our method neither requires 3D data nor prior knowledge on camera pose distribution during training. Additionally, it utilizes pre-trained models available online for 3D-aware image synthesis, thus further enhances its flexibility and practicality.

Key to our method is the integration of image inverse warping loss and consistency regularization strategies, which is shown to enhance the quality of generated images via latent refinement. We address the challenge of maintaining multi-view consistency by incorporating self-calibrated depth and pose estimation, supplementing 3D information to training of our latent refiner. We realize our latent refiner with a diffusion model, which is capable of maintaining high consistency through a close form of forward-backward consistency regulation. The model is trained to learn geometric consistency in a multi-view setting, guided by a photometric loss and pose matrices. Such a training process ensures the fidelity of generated data across a wide range of views without relying on extensive datasets and heavy computational burden.

We evaluate our method on the FFHQ dataset (Karras et al., 2019) and SDIP Elephants dataset (Mokady et al., 2022). Experimental results show that our method outperforms state-of-the-art 3D-aware GANs in wide-angle conditions in terms of both multi-view consistency and image quality.

# 2 RELATED WORK

**GAN-based image manipulation.** Several studies (Shen et al., 2020; Shen & Zhou, 2021; Härkönen et al., 2020; Wu et al., 2021; Collins et al., 2020; Ling et al., 2021; Deng et al., 2020; Tewari et al., 2020; Zhu et al., 2023) have found that the latent space of the StyleGAN (Karras et al., 2019; 2020b;a; 2021) includes remarkably disentangling attributes and rich semantic information, showcasing the capability to alter images produced by pre-trained GANs via arithmetic manipulations of their latent

vectors. These alterations keep generated images within their original distribution while infusing them with new attributes. By applying pose disentanglement arithmetic manipulations to pre-trained models, one can obtain a variety of multi-view face generators. This inspires us to explore the conversion of a 2D GAN into a 3D-aware GAN. However, manipulations of high-dimensional vectors in pose generation make other face attributions (e.g., geometry) inconsistent across viewpoints. Also, latent semantic information is sensitive, and thus different samples may lead to different generation results. Several works (Tewari et al., 2020) require additional labeling, pre-trained attributes classifier, or 3D reconstruction to obtain accurate 3D parameters. DragGAN (Pan et al., 2023) learned latent codes through a loss function that enforces the movement of user-selected image locations towards target ones. Unlike previous methods, our method learns view-consistent latent codes using pose information through self-calibrated learning.

**3D-aware GANs.** The success of neural radiance fields (NeRF) (Mildenhall et al., 2021) in multi-view rendering has opened a new research direction in 3D-aware GANs (Chan et al., 2022; Skorokhodov et al., 2022; Schwarz et al., 2022; Gu et al., 2022; Deng et al., 2022; Zhao et al., 2022). In particular, EG3D (Chan et al., 2022) learned a tri-plane representation with upsampling layers. GRAM (Deng et al., 2022) learned a surface manifold representation with implicit isosurfaces, like the multi-plane representation used in GMPI (Zhao et al., 2022). EpiGRAF (Skorokhodov et al., 2022) conditioned a discriminator with camera pose data to reduce the complexity of its architecture, improving the training time. Compared with methods that manipulate the latent space, 3D-aware GANs achieve better view consistency. However, existing 3D-aware GANs require large-scale datasets to maintain the generation quality (e.g., high resolutions). Moreover, it is hard to reuse pre-trained models to build 3D-aware GANs. Recently, Ray-conditioning (Chen et al., 2023) utilized a 2D GAN conditioned by a light field prior for multi-view rendering. Various inversion and editing techniques can be created by applying StyleGAN, without the use of a geometry 3D prior. However, directly use of the latent space of the StyleGAN may not preserve the identity of objects across views. Our method can maintain the identity of objects in multiple views due to the aid of depth and pose information, which are learned efficiently with self-supervision. In addition to the methods we have discussed, there is an ongoing research trend adapting these methods for handling large-scale, complex data (Sargent et al., 2023; Skorokhodov et al., 2023; Shi et al., 2022), as well as exploring their integration with diffusion models (Xiang et al., 2023; Yang et al., 2023; Tseng et al., 2023) and text-based techniques (Seo et al., 2023; Rombach et al., 2022; Qian et al., 2024; Liu et al., 2023b; 2024; Long et al., 2023). However, such an exploration is beyond the scope of our work.

## 3 Proposed Method

### 3.1 Overview

We aim to design a method that can generate high-quality and multi-view consistent images of a given subject, e.g., a human face. This objective could be achieved by using image-to-image translation techniques (Isola et al., 2017; Zhu et al., 2017). However, training of a conditional image-to-image translation model is known to be computationally expensive while generation results are limited in both image resolution and viewpoints. In this paper, instead of working directly on the image domain, we propose to manipulate the latent space of a GAN model. Specifically, we adopt a pre-trained 2D GAN (e.g., StyleGAN2 (Karras et al., 2020b)), denoted as $G$, to generate images of a subject defined by a latent code $w$ in different viewpoints. To do so, a straightforward approach is to generate latent vectors $\hat{w}$ by sampling the latent space $W$ (e.g., the GANSpace (Härkönen et al., 2020)), i.e., $\hat{w} \sim p_w, w \in W$. However, we observed that such an approach may result in view-inconsistent images (see Fig. 1). To address this issue, we develop a latent refiner that revises sampled codes $\hat{w}$ into view-consistent latent codes $\hat{w}_+$ by taking into account the geometric information (depth and pose) of the input subject across views. We realize our latent refiner by a guided diffusion model (Dhariwal & Nichol, 2021). We learn the depth and pose information in a self-supervised learning fashion. The pipeline of our method is illustrated in fig. 2, where each step is described in a respective subsection below.

### 3.2 Pseudo latent-image generation

We adopt GANSpace (Härkönen et al., 2020) to construct the latent space $W$ on which latent codes $w$ are sampled. We consider the sampled latent codes $w$ and their corresponding images $G(w)$ generated

Figure 2: Pipeline of our method. Our method consists of three steps: 1) pseudo image-latent generation, 2) depth and pose estimation, and 3) latent refining. At each step, a corresponding multi-view dataset is constructed and utilized to train respective models.

by a pre-trained GAN $G$ as pseudo latent-image ground truth and use them to train other modules such as depth estimator, pose estimator, and latent refiner. GANSpace is selected due to its support in navigating the latent space in an unsupervised manner. Specifically, given a latent code $w$, we repeatedly rotate it to produce a sequence of $\hat{w}$ covering a range of viewpoints of the same subject as,

$$\hat{\mathbf{w}} = \mathbf{w} + \mathbf{V}\mathbf{x}$$
$$= \{\hat{w}_1, \hat{w}_2, \hat{w}_3, \cdots, w, \cdots, \hat{w}_{n-2}, \hat{w}_{n-1}, \hat{w}_n\}, \quad (1)$$

where $\mathbf{V}$ is a matrix of principal directions identified through a PCA procedure in the latent space and $\mathbf{x}$ is the extent of movement.

It should be emphasized that the preliminary set of multi-view images $G(\hat{w})$ exhibits significant inconsistencies. For example, as shown in the 1st and 3rd rows in Fig. 1, we observed artefacts and distortions in the generated images. Despite such imperfection, this initial set of sampled latent codes and their corresponding images provides a reasonable starting point for further processing steps (depth estimator, pose estimator, and latent refiner).

### 3.3 PSEUDO DEPTH AND POSE ESTIMATION

Given the sampled latent codes $w$ and $\hat{w}$, and their corresponding images $G(w)$ and $G(\hat{w})$, we train a depth and a pose estimator. Specifically, we train from scratch an unsupervised depth estimation model based on Monodepth2 (Godard et al., 2019) on our generated data, i.e., $G(w)$ and $G(\hat{w})$. The depth estimation model receives inputs as an image $G(w)$ and its subsequent image $G(\hat{w})$, and estimates a pseudo depth map $d_w$ for the subject captured by the image pair $(G(w), G(\hat{w}))$. Fig. 3 illustrates several pseudo depth estimation results. We noticed disruptive effects at the subject's boundaries. This is because the depth estimator is trained with pseudo data. However, the pseudo depth maps are sufficient to carry out the overall 3D information for further processing in the pipeline.

Similarly, we customize Monodepth2 (Godard et al., 2019) to make it our pose estimator. We train the pose estimator to predict a pose transformation $T_{w \to \hat{w}}$ (a $3 \times 3$ matrix) that transforms the pose from $G(w)$ to that in $G(\hat{w})$. Note that, as shown in Fig. 3, the training data for both the depth estimator and pose estimator is created at uniform intervals from the pseudo-multi-view images $G(w)$ and $G(\hat{w})$.

### 3.4 LATENT REFINEMENT

Let $D_\theta$, with learnable parameters $\theta$, be our latent refiner. We build $D_\theta$ upon the guided diffusion model in (Dhariwal & Nichol, 2021). In particular, $D_\theta$ takes as inputs a latent code $w$, its rotated version $\hat{w}$ achieved by some latent manipulation (Härkönen et al., 2020) with some added noise, a

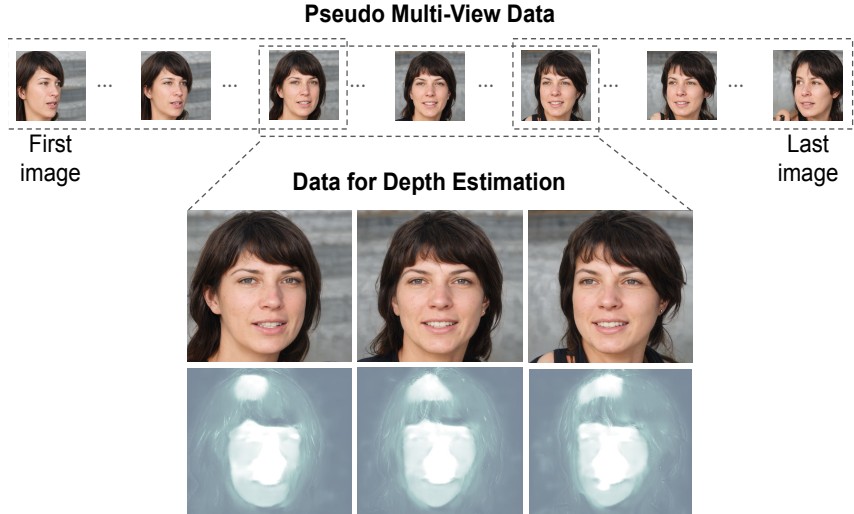

Figure 3: Pseudo depth estimation. Note that the estimated depth has some artifacts, but it still maintains a certain level of consistency from a multi-view perspective.

pose transformation $T_{w \to \hat{w}}$ estimated by the pose estimator, and a time-step variable $t$ (used in the diffusion process). $D_\theta$ aims to rectify $\hat{w}$ (in the latent space) and returns $\hat{w}_+$, a refined version of $\hat{w}$, with better view-consistency:

$$\hat{w}_+ = D_\theta(w, \hat{w}_t, T_{w \to \hat{w}}, t). \tag{2}$$

Subsequently, the refined latent representation $\hat{w}_+$ is used to create an image $G(\hat{w}_+)$ by leveraging the generation ability of the pre-trained GAN $G$. Compared with $G(\hat{w})$, $G(\hat{w}_+)$ shows better quality in terms of photo-realism and subject identity. However, we still found misalignment between $G(\hat{w}_+)$ and $G(w)$. To alleviate this issue, we adopt the warping method presented in (Godard et al., 2019), to warp $G(\hat{w}_+)$ using both the estimated depth map $d_w$ and pose transformation $T_{w \to \hat{w}}$ towards the camera view of $G(w)$. This step results in view-consistent images $\text{Warp}(G(\hat{w}_+), d_w, T_{w \to \hat{w}})$,

$$\text{Warp}(G(\hat{w}_+), d_w, T_{w \to \hat{w}}) \sim K T_{w \to \hat{w}} d_w K^{-1} G(w), \tag{3}$$

where $K$ is an intrinsic calibration matrix.

## 3.5 TRAINING

We train our entire pipeline (see Fig. 2) using self-supervised approach. Due to the absence of real ground truth $\hat{w}$, we are unable to employ a standard denoising diffusion loss as in (Dhariwal & Nichol, 2021). Therefore, we introduce an inverse warping loss that utilizes pseudo depth and pose information to constrain the image generation process and "$x_0$-formulation" (Salimans & Ho, 2022; Karnewar et al., 2023), i.e., training of the diffusion model at 0-th time-step. Specifically, our inverse warping loss is defined as:

$$\mathcal{L}_{warp} = \|\text{Warp}(G(\hat{w}_+), d_w, T_{w \to \hat{w}}) - G(w)\|^2, \tag{4}$$

where $\text{Warp}(G(\hat{w}_+), d_w, T_{\hat{w}})$ is the warping result of $G(\hat{w}_+)$ using both pseudo depth map $d_w$ and pseudo pose transformation $T_{w \to \hat{w}}$, defined in Eq. (3).

We further design a regularization loss $L_{reg}$ to constrain the deviation of the refined latent code $\hat{w}_+$ from its original code $w$, and the consistency between corresponding visual attributes encoded in these latent codes as follows:

$$\mathcal{L}_{reg} = \mathcal{L}_{latent} + \lambda_{feat} \mathcal{L}_{feat}, \tag{5}$$

$$\mathcal{L}_{latent} = \|\hat{w}_+ - w\|^2, \tag{6}$$

$$\mathcal{L}_{feat} = \|F_G(\hat{w}_+) - F_G(w)\|^2, \tag{7}$$

where $F_G$ represents an intermediate layer in the generator of $G$, and we set $\lambda_{feat} = 1$.

The total loss to train our entire pipeline is defined as

$$\mathcal{L} = \mathcal{L}_{warp} + \lambda_{reg}\mathcal{L}_{reg}, \tag{8}$$

where we set $\lambda_{reg} = 1$.

## 4 EXPERIMENTS

### 4.1 EXPERIMENTAL SETUP

We evaluated our method and related works on the Flickr-Faces-HQ (FFHQ) (Karras et al., 2019) and SDIP Elephants (Mokady et al., 2022) datasets. For the related works, we evaluated existing 3D-aware GANs, including GMPI (Zhao et al., 2022), Ray-conditioning (Chen et al., 2023), EpiGRAF (Sko-rokhodov et al., 2022), and 2D latent manipulation methods, including GANSpace (Härkönen et al., 2020), DragGAN (Pan et al., 2023). We also compared our work with state-of-the-art diffusion models for image-to-3D generation using SyncDreamer (Liu et al., 2024). Although this model is trained in a multi-view setting, which differs from our setup that generates 3D-aware images from 2D inputs, we included it for quality comparison.

We adopted the pre-trained StylegGAN2 ($512 \times 512$ resolution) in (Karras et al., 2020b) to implement 2D latent manipulation methods and ours. We used a truncated range of 0.8 to generate a test set (including 6,000 images) for all experimented methods.

We utilized the GANspace (Härkönen et al., 2020) with a manipulation range of $[-3, 3]$ to initialize our latent space. We generated 800 and 600 samples and created 30 pairs of multi-view images for each face and elephant subject to train our latent refiner. We used 200 pair sets to train Monodepth2 (Godard et al., 2019) for both depth estimation and pose estimation. All experiments were carried out on an NVIDIA GeForce RTX 3090 with 24GB memory. To obtain the animal pose for 3D aware GAN models in evaluation, we used the pose estimator model (Ye et al., 2022).

### 4.2 MULTI-VIEW GENERATION

For quantitative evaluation, we employed the Frechet Inception Distance (FID) (Heusel et al., 2017) and Kernel Inception Distance (KID) (Bińkowski et al., 2018) as performance metrics. These metrics are widely used for assessment of the perceptual quality of generated images against real images (i.e., photo realism). However, unlike previous research, we first evaluated our method for multi-view consistent image generation by using a pose estimator model to obtain the overall distribution of poses. We divided this distribution into several zones and compared real and generated samples within each zone. We then generated and selected samples from each zone, resulting in a total of 6,000 images for evaluation. By comparing images of the same instances from multiple viewpoints within these zones, we assessed the multi-view consistency and image quality of generated images. This experimental design, grounded in the dataset's angle distribution, enables a precise and substantial evaluation of our method's ability to produce high-quality and view-consistent images.

We also compared our method with existing ones using the Identity Consistency (IC) metric (Tov et al., 2021; Deng et al., 2019) and Structural Similarity Index (SSIM) (Wang et al., 2004). IC and SSIM reflect the consistency of a generated image against its original input. Specifically, we measured IC by utilizing face recognition features (Tov et al., 2021; Deng et al., 2019) for faces and CLIP features (Radford et al., 2021) for elephants.

**Quantitative results.** We report the performance of our method and existing methods in Table 1, 2. Experimental results indicate that our method not only produces images of high fidelity from multiple viewpoints but also maintains the identity and structure of the original subject across viewpoints. As shown in Table 1, our method achieves the best performance in terms of the photo-realism of generation results using both the FID and KID scores. For consistency assessment, our method ranks first on the SSIM score and second on the IC score. Compared with EpiGRAF (Skorokhodov et al., 2022) (the first-ranked method in terms of the IC score), our method achieves a comparable IC score but a significantly better SSIM score. Table 2 also confirms the superiority of our method over existing ones, proven by our achieved lowest FID, KID and IC scores. We also report the training

Table 1: Quantitative evaluations on FFHQ. Photo-realism is measured using the FID and KID metrics (lower scores mean better performances). Consistency is measured using the IC and SSIM metrics (higher scores mean better performances). Training efficiency is measured as the training time required to generate a 512x512-pixel image. Best performances are highlighted.

| FFHQ | FID | KID | $IC_{Face}$ | SSIM | Training time |
|---|---|---|---|---|---|
| GANspace | 48.98 | $1.145 \times 10^{-2}$ | 0.958 | 0.484 | 0.5 hour |
| DragGAN | 68.45 | $2.231 \times 10^{-2}$ | 0.895 | 0.497 | - |
| GMPI | 54.95 | $2.199 \times 10^{-2}$ | 0.971 | 0.509 | 40 hours |
| Ray Cond | 49.41 | $1.547 \times 10^{-2}$ | 0.960 | 0.433 | 27 hours |
| EpiGRAF | 57.78 | $1.953 \times 10^{-2}$ | **0.975** | 0.454 | 24 days |
| SyncDreamer | 72.68 | $7.102 \times 10^{-2}$ | 0.890 | 0.555 | 18 days |
| Ours | **48.25** | **1.137** $\times 10^{-2}$ | 0.972 | **0.581** | 18 hours |

Table 2: Quantitative evaluations on SDIP Elephants. Like Table 1, the same performance metrics are used here.

| SDIP Elephants | FID | KID | $IC_{Clip}$ | SSIM | Training time |
|---|---|---|---|---|---|
| GANspace | 67.32 | $2.380 \times 10^{-2}$ | 0.915 | 0.459 | 0.5 hour |
| DragGAN | 70.38 | $2.573 \times 10^{-2}$ | 0.850 | 0.481 | - |
| GMPI | 78.12 | $4.632 \times 10^{-2}$ | 0.764 | 0.317 | 30 hour |
| Ray Cond | 69.32 | $2.475 \times 10^{-2}$ | 0.877 | 0.503 | 1 day |
| EpiGRAF | 75.32 | $5.632 \times 10^{-2}$ | 0.792 | 0.326 | 10 days |
| SyncDreamer | 69.89 | $6.473 \times 10^{-2}$ | 0.8113 | **0.602** | 18 days |
| Ours | **65.78** | **2.294** $\times 10^{-2}$ | **0.932** | 0.571 | 12 hours |

time of all the experimented methods in Table 1, 2. Our training duration is expedient compared with other 3D-aware methods, and while it may be slower than 2D latent space techniques, it effectively balances the image quality and consistency of generated images over an extensive range of angles.

We conducted a detailed evaluation of our method and existing ones by assessing their generation quality on various finer angular ranges including [-20°,20°], [-40°,40°], and full range. We report the results of this experiment in Table 3, 4. As shown in Table 3, our method achieves superior overall performance over the experimented baselines, across all examined rotation angle ranges (column "Full range"). In more detail, within the angular range $[-20°, 20°]$, our model performs on par with GANspace (Härkönen et al., 2020) (the leading baseline in this range), but surpasses all the baselines in the intermediate range $[-40°, 40°]$ on both the FID and KID metrics. Also, Table 4 shows higher IC and SSIM scores achieved by our method in a wide range of angles, demonstrating improved multi-view consistency at wider angles.

**Qualitative results.** We showcase the qualitative results of our method and other methods in Fig. 4. Here, we visually assess the quality of face and elephant images generated by the methods. As observed, our method maintains identity consistency and high-quality generation results. GANspace (Härkönen et al., 2020) exhibits noticeable distortions in facial representation as the angle increases. DragGAN (Pan et al., 2023) shows different semantic attributes or identity changing, revealing the limitations of latent feature matching. Ray-conditioning (Chen et al., 2023) alters attributes, such as the degree of smiling, as the face rotates, and the elephant ears. In EpiGRAF (Skorokhodov et al., 2022) and GMPI (Zhao et al., 2022), attributes like ears and hair become blurry or appear flat with rotation, while only the central facial region retains a three-dimensional appearance. In addition, when trained on SDIP Elephants, significant breakdown in the structure and substantial distortions beyond a certain angle. SyncDreamer (Liu et al., 2024) despite being trained on 3D data, still exhibits incorrect geometry transformations. We also provide a comparison to diffusion-based view synthesis in the supplementary material.

Table 3: Quantitative evaluations for view-consistency using FID and KID metrics. For each method and each experimental setting (i.e., angular range), we report the FID (left) and KID (right) scores.

| FID/KID | -20° to 20° | -40° to 40° | Full Range |
|---|---|---|---|
| GANspace | **47.56** — **1.193** $\times 10^{-2}$ | 46.92 — 1.085 $\times 10^{-2}$ | 48.98 — 1.145 $\times 10^{-2}$ |
| DragGAN | 60.21 — 1.823 $\times 10^{-2}$ | 59.73 — 1.797 $\times 10^{-2}$ | 68.45 — 2.231 $\times 10^{-2}$ |
| GMPI | 49.26 — 1.654 $\times 10^{-2}$ | 52.31 — 1.931 $\times 10^{-2}$ | 54.95 — 2.199 $\times 10^{-2}$ |
| Ray Cond | 51.05 — 1.886 $\times 10^{-2}$ | 48.28 — 1.534 $\times 10^{-2}$ | 49.41 — 1.547 $\times 10^{-2}$ |
| EpiGRAF | 52.59 — 1.768 $\times 10^{-2}$ | 53.93 — 1.665 $\times 10^{-2}$ | 57.78 — 1.953 $\times 10^{-2}$ |
| Ours | 47.87 — 1.195 $\times 10^{-2}$ | **46.28** — **1.079** $\times 10^{-2}$ | **48.25** — **1.137** $\times 10^{-2}$ |

Table 4: Quantitative evaluations for view-consistency using IC and SSIM metrics. For each method and each experimental setting (i.e., angular range), we report the IC (left) and SSIM (right) scores. Higher scores mean better performances. Best performances are highlighted.

| Method | -20° to 20° | -40° to 40° | Full Range |
|---|---|---|---|
| GANspace | 0.984 — 0.583 | 0.958 — 0.484 | 0.958 — 0.484 |
| DragGAN | 0.945 — 0.697 | 0.925 — 0.587 | 0.895 — 0.497 |
| GMPI | 0.988 — 0.522 | 0.980 — 0.516 | 0.971 — 0.509 |
| Ray Cond | 0.985 — 0.495 | 0.972 — 0.451 | 0.960 — 0.433 |
| EpiGRAF | 0.991 — 0.494 | **0.982** — 0.463 | **0.975** — 0.454 |
| Ours | **0.992** — **0.73** | 0.972 — **0.59** | 0.972 — **0.581** |

### 4.3 OTHER PRE-TRAINED 2D GANS

Our method can work with different pre-trained GANs. In particular, we can synthesize multi-view images using pre-trained 2D GANs trained on different public datasets including SD-LSUN-Elephant, Giraffe, Parrot (Mokady et al., 2022) and Metfaces (Karras et al., 2020a). Our method therefore eliminates need for specialized and large-scale datasets, and costly generative model training. We can also generate a multi-view dataset using pre-trained 2D GAN models that are readily accessible online. We prove this ability in Fig. 5.

### 4.4 3D RECONSTRUCTION

We visually compare our method and other ones in the application of 3D reconstruction in Fig. 6, 7. The meshes from GMPI (Zhao et al., 2022) and EpiGRAF (Skorokhodov et al., 2022) are created by a marching cubes algorithm. For other 3D reconstruction methods, we use COLMAP (Schönberger & Frahm, 2016) to generate dense 3D point clouds from 30 synthesized images.

As shown in Fig. 6, 7, compared with other point cloud reconstruction methods, e.g., Ray conditioning (Chen et al., 2023), DragGAN (Pan et al., 2023), GANspace (Härkönen et al., 2020), our method shows more accurate facial reconstructions. Compared with mesh generation methods, e.g., GMPI (Zhao et al., 2022), EpiGRAF (Skorokhodov et al., 2022), although there are some noisy points in our results, it is evident that our method can capture more details, such as the ears and the side profile of the faces. Especially, the results show that while GANspace-generated images appear with relatively smooth multi-view consistency, the reconstructions often focus only on the nose area. Even when multi-view images created by GANspace result in successful reconstructions, our method's generated multi-view images showcase more detailed reconstructions, particularly capturing finer details like the pupils more precisely. Conversely, our method results in more comprehensive reconstructions, capturing the overall facial structure with greater fidelity. This distinction underscores the effectiveness of our method in maintaining geometric consistency across multiple viewpoints.

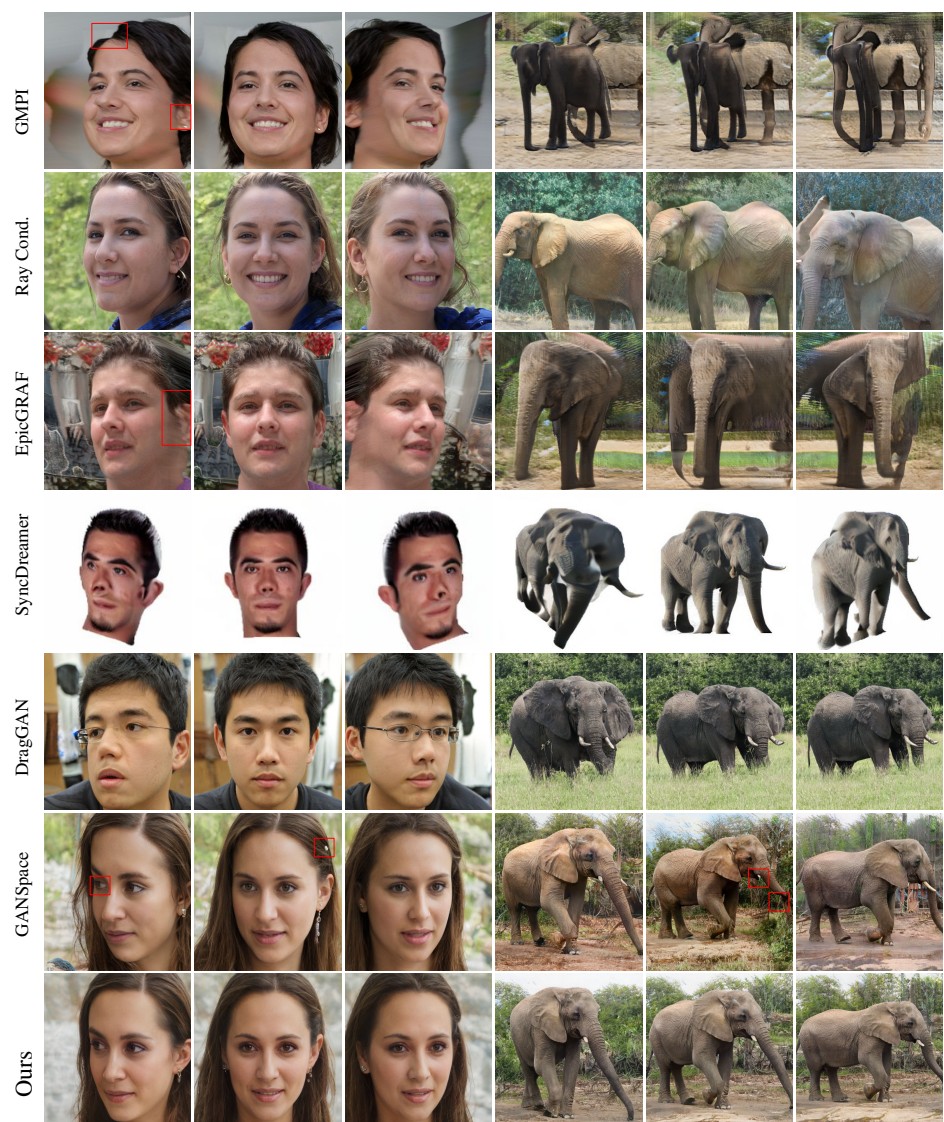

Figure 4: Comparison of our method with existing works. Our approach ensures photo-realistic image quality and the preservation of multi-view consistency including identity, geometry, and appearance characteristics, irrespective of viewing angles.

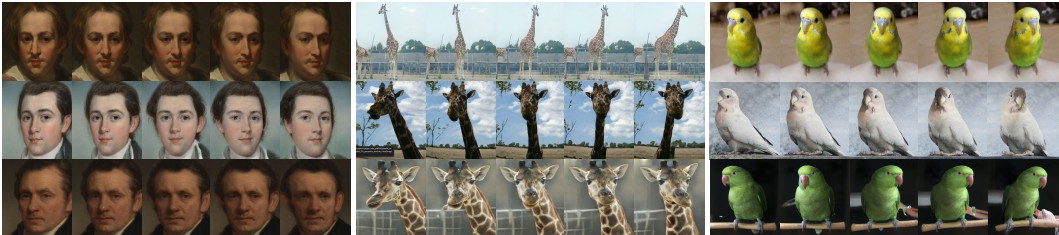

Figure 5: Generation results with Metfaces, SD-LSUN-Elephant, Giraffe, Parrot pre-trained GANs.

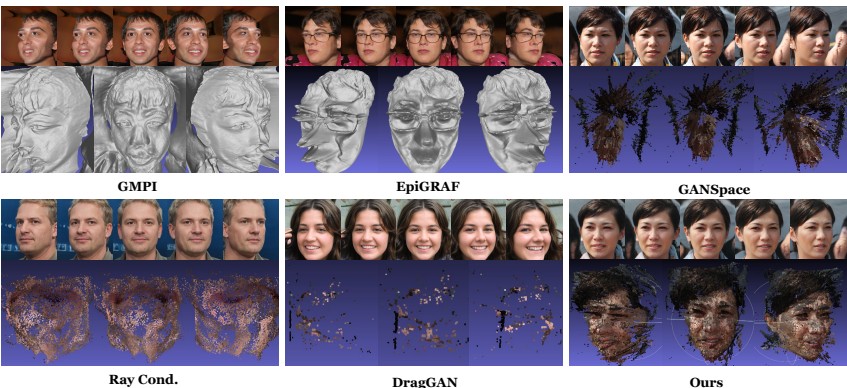

Figure 6: Comparison of our method and existing ones in 3D reconstruction.

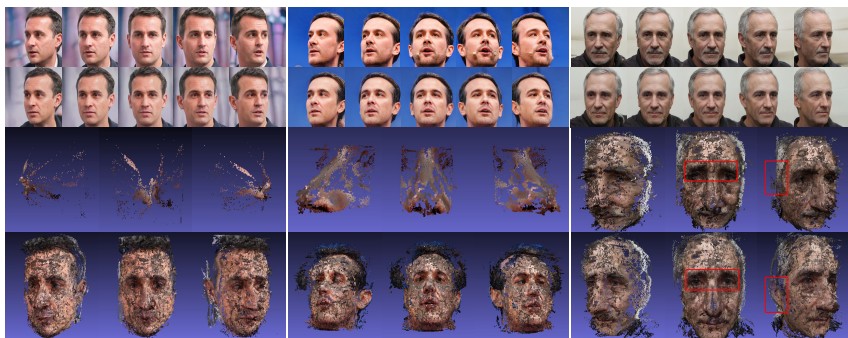

Figure 7: Qualitative evaluation of 3D reconstruction by GANspace (Härkönen et al., 2020) vs. ours. The first row of multi-view images is generated through GANspace, while the samples below are from our method. In the 3D reconstructions, the upper rows are created using GANspace, and the bottoms are produced from our samples.

### 4.5 LIMITATIONS

Our method is not without limitations. In particular, the performance of our method relies on the initial set of multi-view latent codes used to construct the latent space for subsequent processing. Our method thus requires the latent manipulations to be able to generate reasonable results. Hence, it is challenging to construct a meaningful latent space if pose disentanglement is poorly executed, preventing the creation of initial multi-view images with reasonable quality.

## 5 CONCLUSION

We propose a novel method for multi-view image generation. Our method first generates pseudo latent-image samples from a latent space using a pre-trained 2D GAN with latent manipulation ability. The generated pseudo latent-image samples are used to train a depth estimator and a pose estimator, which are then employed to condition a latent refinement process governed by a diffusion model. By introducing new warping and regularization losses, our method can be trained using self-supervised approach. We demonstrated the robustness of our method via extensive experiments on benchmark datasets. Experimental results show that our method achieves the best geometric and semantic feature consistency in a wide rotation margin.

The creation of multi-view representations used to initialize a latent space in our method relies on the effectiveness of a 2D latent manipulation method, such as the one offered by GANspace Härkönen et al. (2020). This means the necessity for further advancements to improve the capacity of latent manipulation. Extending the proposed method to incorporate other generative models, such as diffusion models, would also be a potential research direction.

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

Table 5: Evaluation of the loss functions

| Loss function | FID | KID | IC | SSIM |
|---|---|---|---|---|
| w/o $\mathcal{L}_{latent}$ | 54.25 | $1.713 \times 10^{-2}$ | 0.843 | 0.482 |
| w/o $\mathcal{L}_{feature}$ | 48.73 | $1.094 \times 10^{-2}$ | 0.754 | 0.454 |
| w/o $\mathcal{L}_{warping}$ | 101.37 | $4.137 \times 10^{-1}$ | 0.232 | 0.143 |
| Full loss | 48.25 | $1.137 \times 10^{-2}$ | 0.972 | 0.581 |

## A  APPENDIX

In this supplementary document, we conduct additional quantitative and qualitative studies. In particular, we verify the effect of loss functions in Section A.1 and provide a comparison to diffusion-based novel view synthesis in Section A.2. We showcase another ability of our method in enabling multi-view consistent image editing in Section A.3. We present additional results of multi-view image synthesis.

### A.1  LOSS FUNCTIONS STUDY

In this section, we study the effect of the loss functions used in our method. In particular, we remove the warping loss and some parts of the regularization losses, including the feature loss and the latent loss, and re-train the corresponding models for comparison. We use the Frechet inception distance (FID) (Heusel et al., 2017), Kernel inception distance (KID) (Bińkowski et al., 2018), IC, and SSIM as performance metrics for the quality and consistency of multi-view image synthesis.

We report the results of these experiments in Table 5. The results clearly illustrate the significant influence of the warping loss, particularly regarding the image quality and view-consistency of generated images. The integration of feature and latent losses plays a pivotal role in refining latent codes, making them aligned with near-source view latent codes. We visualize several generation results by the loss functions in Fig. 8. As shown, without feature and latent regularizations, although still appearing similar to the source view, there are differences in the fine details of the generation results such as the extent to which the hair covers the ears or smile strength.

### A.2  COMPARISON TO DIFFUSION-BASED NOVEL VIEW SYNTHESIS

We included a comparison of our method with diffusion-based methods in  9. In this comparison, we used the center images from one of our samples as the input for image-to-3D conversion. Some existing methods predict multi-view images at specific angles, so we could not match the angles exactly. However, this does not affect image quality and multi-view consistency comparison. The results show that although diffusion-based methods can cover broader angles and a variety of objects, they often fall short in photorealism and identity consistency with the input image and across different views. In contrast, our method was preserved well. This demonstrates that our approach is useful for efficiently obtaining 3D-aware images, even for specific datasets that are difficult to generalize.

Our method operates without the need for 3D data or camera pose information, whereas diffusion-based methods require abundant 3D data for fine-tuning. Despite this significant difference, our method performs comparably by utilizing a self-calibration mechanism that estimates and adjusts conditions based on limited initial information. Additionally, our approach employs a lightweight training strategy, as it does not require fine-tuning of a GAN. Despite these fundamental differences, our method demonstrates strong performance in generating high-quality and consistent multi-view images, as seen in our comparisons with diffusion-based novel view synthesis.

### A.3  MULTI-VIEW CONSISTENT IMAGE EDITING

In this section, we showcase another application of our method when integrated with image editing techniques. In particular, we adopt InterfaceGAN (Shen et al., 2020) as an image editing technique to edit attributes of faces generated by our method. We illustrate several results of this application in Fig. 10.

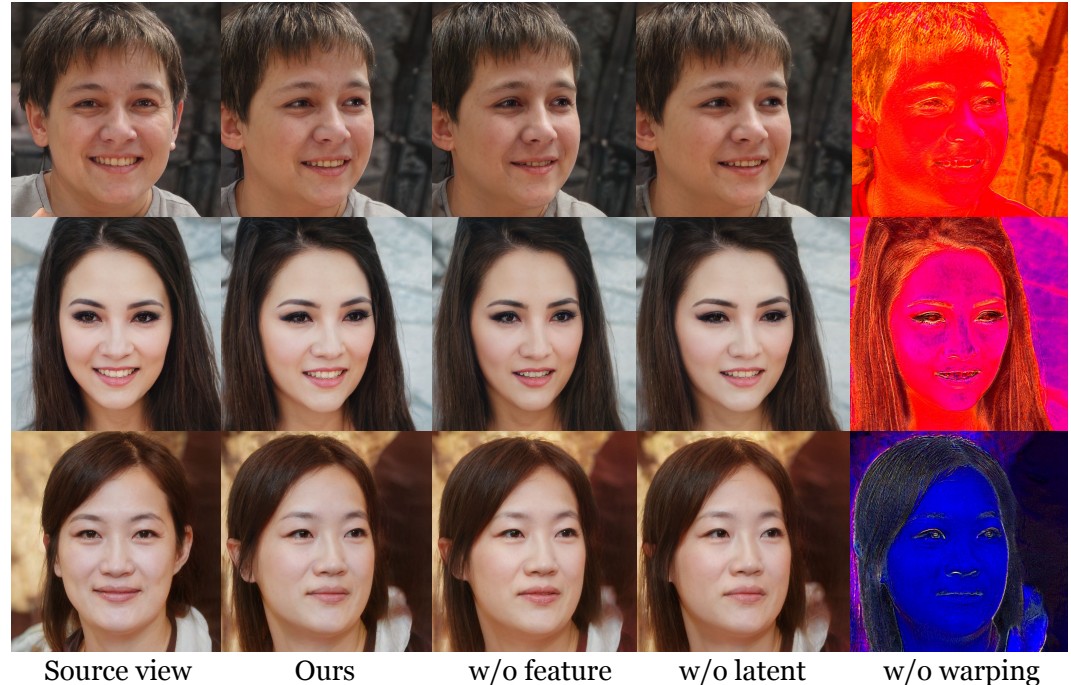

| Source view | Ours | w/o feature | w/o latent | w/o warping |

Figure 8: Visualization of the effects of the loss functions to generation results.

The results show that our method, with the ability to generate view-consistent images, can provide high quality inputs to InterfaceGAN, enabling multi-view consistent image editing. This also shows the potential of our method in facilitating the exploration of a diverse range of image editing techniques. In Fig 10, the last row is edited results with inputs generated by GANspace (Härkönen et al., 2020). Compared with our method, it can be observed that the hairstyle and the person's identity generated by GANspace is less view-consistent.

### A.4 MULTI-VIEW IMAGE SYNTHESIS

In this section, we provide additional qualitative results showcasing the ability of our method to generate facial images under various viewpoints (see Fig 11, 12). These results show that our method can maintain the view consistency in generated faces across a wide range of angles and expressions.

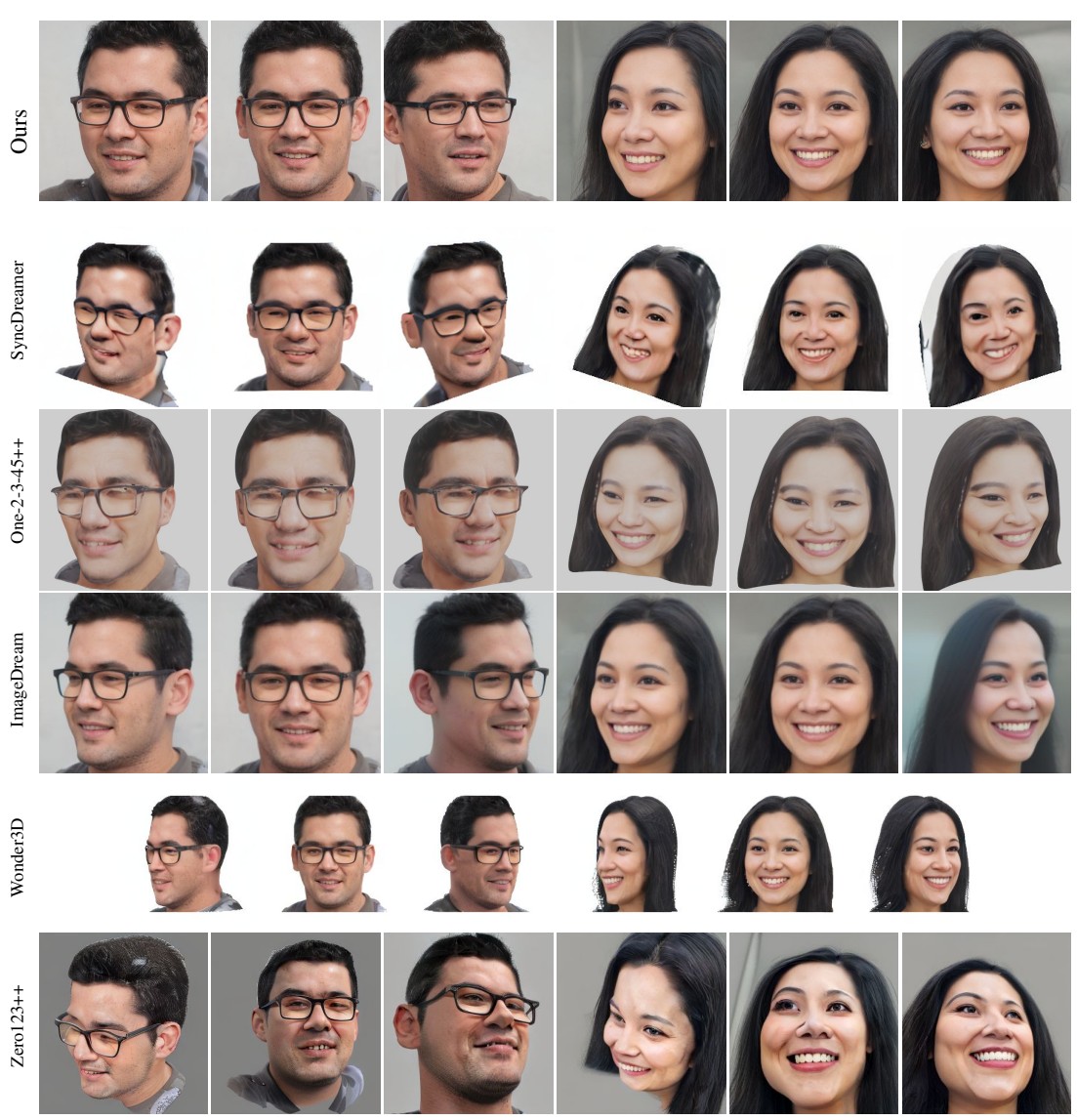

Figure 9: Comparison of our method with latest novel view synthesis Liu et al. (2024; 2023a); Long et al. (2023); Wang & Shi (2023); Shi et al. (2023). While our approach covers a limited range of angles, it ensures photo-realistic image quality and the preservation of multi-view consistency, including identity, geometry, and appearance characteristics. Although the latest novel view synthesis-based 3D generation models can cover a broader range of angles, they often fall short in realism and exhibit discrepancies in identity when compared to the input images.

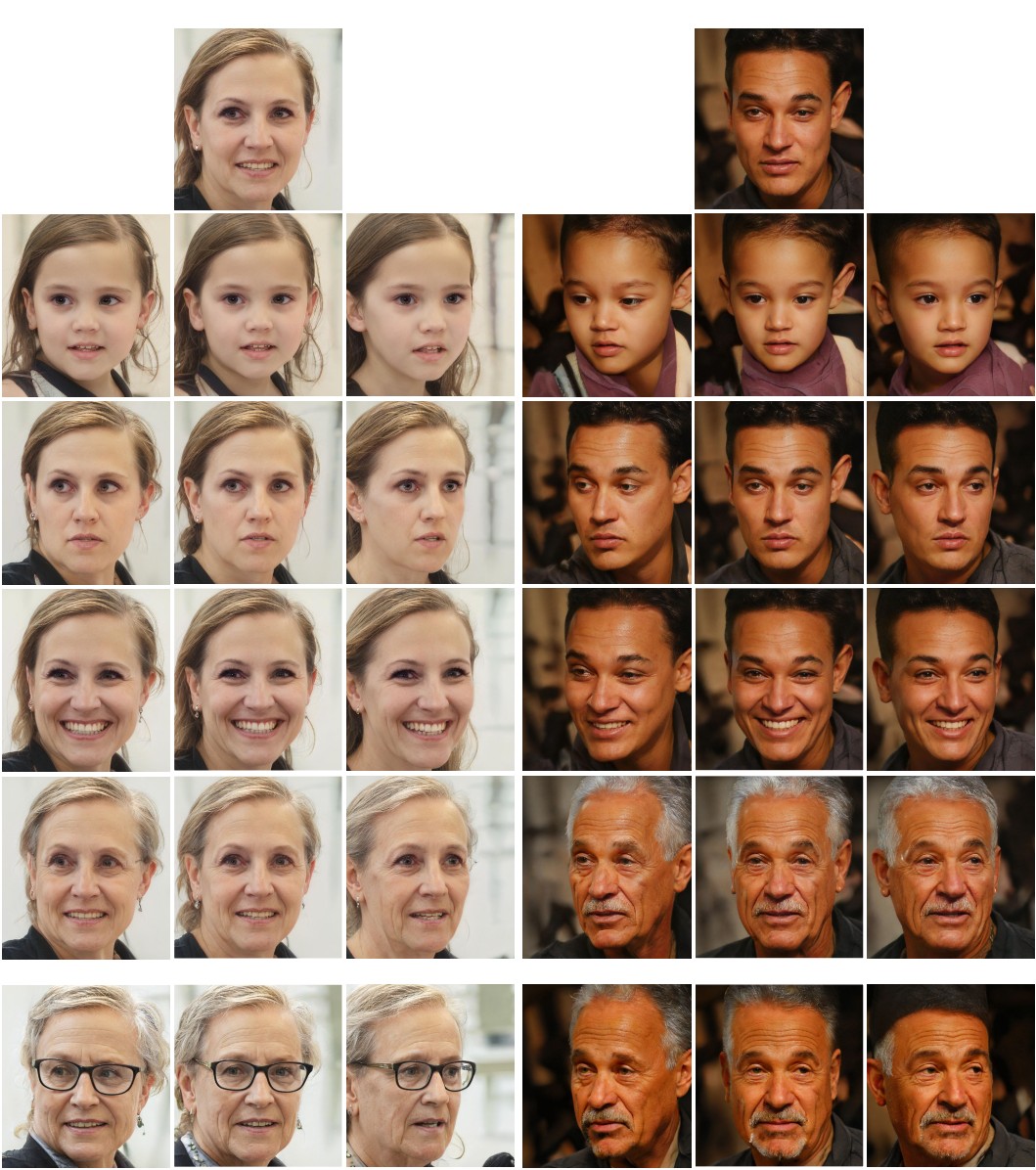

Figure 10: Illustration of multi-view consistent image editing. The first row includes source images, following rows show edited results for different attributes, e.g., age, expression. The last row showcases the results of GANSpace multiview image editing, emphasizing less view consistency, particularly noticeable in aspects like the missing hair, when compared with our method.

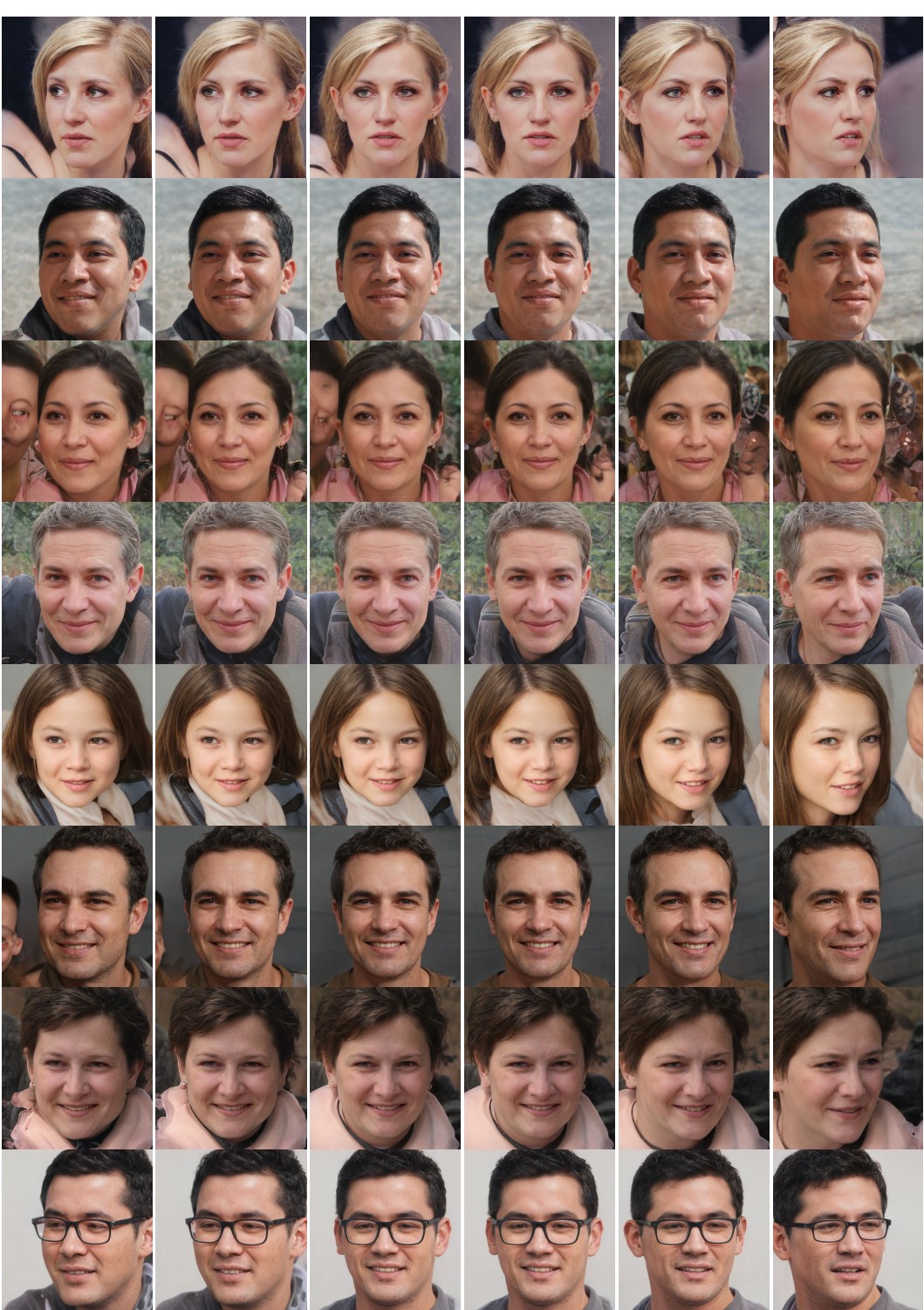

Figure 11: Multi-view facial images generated by our method on the FFHQ dataset (Karras et al., 2019) - part 1.

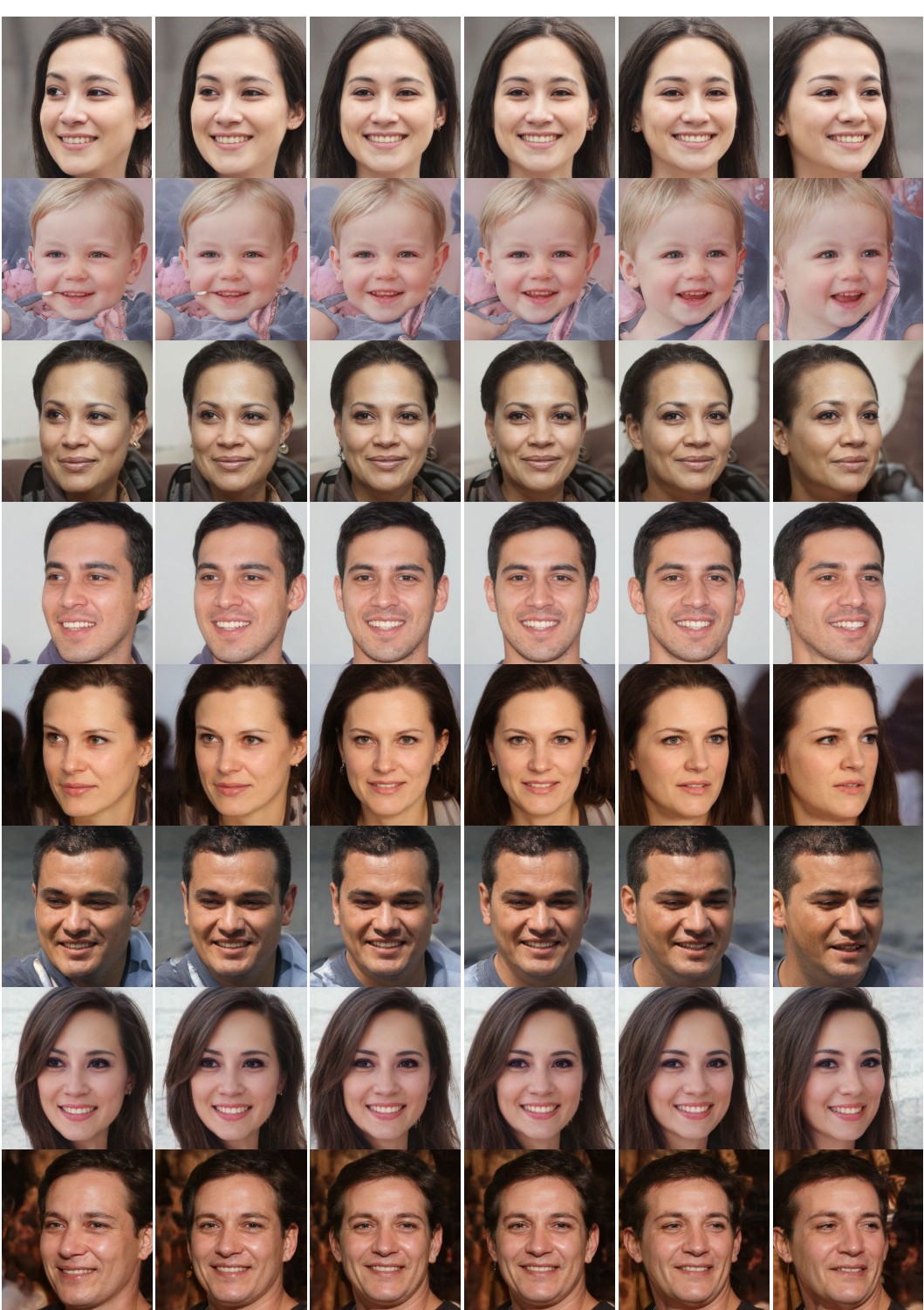

Figure 12: Multi-view facial images generated by our method on the FFHQ dataset (Karras et al., 2019) - part 2.

