# OpenReview forum: "Multi-view Consistent Image Generation through Self-calibrated Latent Refinement"
_ICLR.cc/2025/Conference — ICLR 2025 Conference Withdrawn Submission_

### Official Review · Reviewer_cKyd · 2024-11-03

**Soundness:** 2
**Presentation:** 3
**Contribution:** 3
**Rating:** 5
**Confidence:** 5

**Summary:**

The paper proposes a multi-view aware image generation method using GANs. The method uses diffusion models to refine the latents with image inversion and consistency regularization losses, corresponding to self-calibrated pose and depth information, which is then used in the training of the GAN. The method does not use any explicit 3D or multi-view information. Extensive experiments and comparisons on the faces and elephants datasets validates the method.

**Strengths:**

1. Method - The paper combines the benefits of diffusion models within GANs to solve the problem - this is an interesting direction of work. Additionally, the method is logical and makes sense. With various consistency losses, and guidance from diffusion models along with pose information, multi-view consistency in image generation is achieved.

2. The paper is well written.

3. Comparisons with SOTA methods on the faces and elephants datasets is provided.

**Weaknesses:**

1. Line 79-80: We also choose GANs
instead of diffusion models (Ho et al., 2020) as, compared with diffusion models, by design, GANs’
latent spaces are more interpretable and can be computed in much faster speed.

I don't think this is true. To prove otherwise, I suggest that the authors apply the method on a diffusion backbone instead of GAN backbone.

2. I am curious about the choice of datasets - faces and elephants. It appears very cherry picked. To counter this, I suggest that the authors apply the method on other datasets such as those used by SyncDreamer.

3. For comparisons, was the pretrained model of SyncDreamer used or was it retrained on the faces and elephants datasets?

If it's the pretrained model, then the training time of 18 days makes sense. In that case, the comparison in terms of performance is not fair due to domain gap issues, and training time comparison is not fair due to training on different datasets.

Similarly, please clarify this issue for the other comparison methods as well.

**Questions:**

I have combined the questions with the weaknesses.

---

### Official Review · Reviewer_yCwE · 2024-11-03

**Soundness:** 2
**Presentation:** 1
**Contribution:** 2
**Rating:** 3
**Confidence:** 4

**Summary:**

This paper introduces a novel 3D-aware image generation framework that achieves high-quality and view-consistent image generation using a pre-trained 2D GAN, enhanced by self-calibrated depth and pose estimation. The proposed method leverages the semantic latent space of 2D GANs for multi-view consistency without requiring extensive 3D datasets or prior camera pose knowledge.

**Strengths:**

1. **contribution**：It presents a novel method that adapts pre-trained 2D GANs to generate multi-view consistent 3D-like images, eliminating the need for large-scale 3D datasets and prior camera pose information.

2. **experiments**：Experiments on benchmark datasets (FFHQ, SDIP Elephants) demonstrate that this approach achieves superior image quality and multi-view consistency compared to existing state-of-the-art 3D-aware GANs.

**Weaknesses:**

1. **Lack of the technical details.** The core of the paper appears to be using a latent refiner to maintain 3D consistency of latent codes with improved multi-view consistency. However, the main text lacks specific details on this component and does not explain why guided diffusion was chosen as the refiner.
2. **Lack of experiment results.** For multi-view face generation tasks, 3D-aware GANs are typically evaluated on the AFHQv2-Cat dataset, so it is recommended to include results on this dataset to more comprehensively demonstrate the method’s performance. For other multi-view generation tasks, using only the SDIP Elephants dataset may lack sufficient persuasiveness. Given the comparisons with methods such as SyncDreamer and Zero123, further experiments on similar cases（e.g. objaverse） would help validate the proposed method’s effectiveness, enhancing the generalizability and credibility of the results.
3. **Seems unreasonable results.** Regarding the 3D reconstruction results, there is a concern that the depth obtained by this method, as shown in Fig. 3, is quite poor, which theoretically would result in a subpar point cloud. Therefore, the claim of significant superiority in Section 4.4 appears unreasonable.
4. **Lack of the clarity or explanation of the experiment setting.** There is a concern regarding the comparison results on the FFHQ dataset, as the reported FID for GMPI at a 512 resolution on FFHQ is 8.29, while in the comparison section, it is shown as 54.95. This discrepancy raises questions about the validity of the experimental setup and results in the comparison.

**Questions:**

See weakness

---

### Official Review · Reviewer_66Nm · 2024-11-03

**Soundness:** 2
**Presentation:** 3
**Contribution:** 2
**Rating:** 3
**Confidence:** 3

**Summary:**

This work leverages a pre-trained 2D GAN for multi-view consistent, 3D-aware image generation. Starting with a pretrained 2D GAN, such as GANSpace, the method first rotates a latent code to generate a sequence of latent vectors representing different viewpoints of the same subject. Then, a depth and pose estimation network, based on Monodepth2, is trained from scratch in an unsupervised manner. Finally, a latent refinement network, using a diffusion model, is trained to output refined latents with improved view consistency. Experiments on the FFHQ and SDIP Elephants datasets demonstrate the method’s superiority in quality and view consistency compared to existing approaches.

**Strengths:**

1. The proposed method adapts only 2D GAN models but generates multi-view consistent images without requiring a camera distribution prior or 3D ground-truth data.
2. This paper provides a comprehensive comparison with 2D GAN-based methods, 3D-aware GAN methods, and diffusion models as baselines.
3. The method is simple, using an additional warp loss for supervision, yet it significantly improves multi-view consistency over the basic model (2D GANs with GANSpace). It combines the strengths of 2D-based generative models for high quality and 3D-based generative models for enhanced multi-view consistency.
4. The paper is well-written and easy to follow, with clear and informative figures.

**Weaknesses:**

1. The visual results of baselines like EpiGRAF appear quite different from those in the original paper, which makes it unconvincing that 3D-aware generative models could produce such poor results. For multi-view generation tasks on face or car datasets—where the camera distribution can be accurately computed—3D-aware models like EpiGRAF and EG3D, which use 3D representations, should perform well. Why, then, does the proposed method, based only on warp-based loss, look superior in the qualitative results?
2. The setting of the paper is multi-view consistent generation without requiring 3D data or a camera distribution prior. There are closely related 3D-aware generative models without this prior, such as [1], which uses DINO features trained unsupervised on 2D data. A comparison with such models should be added to demonstrate the benefits of the proposed method.
3. The dataset and reconstruction results in this paper focus primarily on faces, which follow a specific camera distribution; 3D-aware generative models (like EG3D) already provide effective solutions in this context. The advantage of the proposed method, which does not require a camera distribution prior, could be better demonstrated on more complex datasets, such as human figures.
4. The “Training Time” reported in Table 1 seems quite unfair. The full training time—including both the training of the base 2D GAN model and the training of the proposed components—should be reported.

Reference:

[1] Learning 3D-Aware GANs from Unposed Images with Template Feature Field

**Questions:**

1.	Why are the visual results for EpiGRAF so poor and different from those in the original paper?
2.	Multi-view consistency is best demonstrated with a free-view video. Does the method produce more consistent video results than EpiGRAF?

---

### Official Review · Reviewer_nB9g · 2024-11-04

**Soundness:** 2
**Presentation:** 2
**Contribution:** 3
**Rating:** 3
**Confidence:** 4

**Summary:**

The paper proposes a 3D-consistent image generation framework. Unlike previous methods that generate inconsistent images from different angles, the proposed method uses a latent refiner with multi-view and geometric preservation capabilities, leveraging depth and pose warping to achieve high consistency in generation from different views. Compared to GANSpace, which causes view inconsistency due to direct manipulation, the proposed method shows high consistency in visual results across different views and performs well in various comparisons and quantitative evaluations such as FID and KID.

**Strengths:**

1. The paper proposes a latent refiner with multi-view and geometric preservation capabilities, achieving high 3D consistency in generation.
2. The proposed method shows good visual results and performs well in quantitative evaluations such as FID and KID. Unlike GANSpace, which results in inconsistent textures and brightness across different views, the proposed method avoids this issue.

**Weaknesses:**

1. In Equation 3, the input on the left side is w+, while on the right side, it is w, which seems to be a mistake. Additionally, the text above Equation 3 mentions warping G(w+), which is inconsistent with Equation 3 and the pipeline in Figure 2.
2. The abstract states that LATENT REFINEMENT provides better view consistency, but there are no ablation experiments to prove this. This is a key contribution of the paper, and there lack of sufficient experiments to demonstrate the effectiveness of the LATENT REFINEMENT module.
3. The method section of the paper is very simple and short. The LATENT REFINEMENT part could be more detailed regarding the guided diffusion model and latent manipulation.

**Questions:**

1. In Figure 2, most images seem to correspond to the actual ones, but the image d_w after depth estimation appears to differ significantly from G(W). Is this the actual effect, or is it just a schematic? This causes some misunderstanding.
2. There are no ablation experiments in the main paper, which are all in the supplementary materials. This seems odd, as ablation experiments can effectively verify the reliability of the method. The authors need to pay attention to this.

---

### Note · Authors · 2024-11-15

I have read and agree with the venue's withdrawal policy on behalf of myself and my co-authors.